# Characterization of a Water-Dispersed Biodegradable Polyurethane-Silk Composite Sponge Using ^13^C Solid-State Nuclear Magnetic Resonance as Coating Material for Silk Vascular Grafts with Small Diameters

**DOI:** 10.3390/molecules26154649

**Published:** 2021-07-31

**Authors:** Takashi Tanaka, Yusuke Ibe, Takaki Jono, Ryo Tanaka, Akira Naito, Tetsuo Asakura

**Affiliations:** 1Department of Veterinary Science, Tokyo University of Agriculture and Technology, Tokyo 183-8509, Japan; bamse.rizea.vicky.polta@gmail.com (T.T.); fu0253@go.tuat.ac.jp (R.T.); 2Polyurethane Research Laboratory, Tosoh Corporation, Mie 510-8540, Japan; yuusuke-ibe-gc@tosoh.co.jp (Y.I.); takaki-jouno-rh@tosoh.co.jp (T.J.); 3Department of Biotechnology, Tokyo University of Agriculture and Technology, Tokyo 184-8588, Japan; naito@ynu.ac.jp

**Keywords:** silk fibroin, small-diameter artificial vascular graft, ^13^C solid-state nuclear magnetic resonance, biodegradable polyurethane

## Abstract

Recently, *Bombyx mori* silk fibroin (SF) has been shown to be a suitable material for vascular prostheses for small arteries. In this study, we developed a softer SF graft by coating water-dispersed biodegradable polyurethane (PU) based on polycaprolactone and an SF composite sponge on the knitted SF vascular graft. Three kinds of ^13^C solid-state nuclear magnetic resonance (NMR), namely carbon-13 (^13^C) cross-polarization/magic angle spinning (MAS), ^13^C dipolar decoupled MAS, and ^13^C refocused insensitive nuclei enhanced by polarization transfer (r-INEPT) NMR, were used to characterize the PU-SF coating sponge. Especially the ^13^C r-INEPT NMR spectrum of water-dispersed biodegradable PU showed that both main components of the non-crystalline domain of PU and amorphous domain of SF were highly mobile in the hydrated state. Then, the small-diameter SF artificial vascular grafts coated with this sponge were evaluated through implantation experiments with rats. The implanted PU-SF-coated SF grafts showed a high patency rate. It was confirmed that the inside of the SF grafts was covered with vascular endothelial cells 4 weeks after implantation. These results showed that the water-dispersed biodegradable PU-SF-coated SF graft created in this study could be a strong candidate for small-diameter artificial vascular graft.

## 1. Introduction

The annual incidence of deaths associated with cardiovascular diseases is predicted to reach 23.3 million worldwide by 2030 [1]. Intravascular surgery has been performed in recent years owing to advances in technology and its low invasiveness to patients; however, owing to problems such as high cost and limitations on the location and size of the affected area, the demand for bypass surgery remains high [2,3]. At present, the commercially available useful artificial vascular grafts are limited to diameters > 6 mm. In cases where a graft with a small diameter of <6 mm is used for bypass surgery in high-demand lower limbs or coronary arteries, commercial artificial vascular graft materials may be occluded by thrombosis or intimal hyperplasia [4,5,6,7,8,9]. Therefore, the development of a small-diameter artificial vascular graft with high patency has been desired.

*Bombyx mori* silk fiber is a natural protein fiber produced by silkworms. It has a long history of use as an excellent textile material [10]. On the other hand, silk fiber has also been used as suture material in the surgical field for a long time [11,12]. Thus, its suitable cytocompatibility, controllable biodegradability, suitable mechanical properties, and minimal inflammatory reaction are suitable for artificial vascular grafts with small diameters [11,12,13]. In fact, many researchers have developed small-diameter silk artificial vascular grafts [13,14,15,16,17]. For example, we have proven that silk fibroin (SF) is suitable for vascular prostheses for small arteries [18]. That is, the patency of a 1.5 mm-diameter SF fiber graft coated with SF was significantly higher than that of an expanded polytetrafluoroethylene (ePTFE) graft used as control (85.1% vs. 30%, *p* < 0.01) at 1 year after implantation in rat abdominal aorta. Sirius red polarization microscopy revealed that the SF content at 2 weeks gradually decreased to 32.9% (*p* < 0.05) at 48 weeks after implantation. On the other hand, the collagen content remarkably increased at 12 weeks (544.9%, *p* < 0.05 vs. 2 weeks). Thus, the important characteristic of small-diameter SF grafts with a high patency ratio is remodeling, which is never observed for commercially available poly(ethylene terephthalate) (PET) fiber or ePTFE grafts. However, the tube of SF fiber grafts coated with SF tends to be rigid; therefore, better transplant results by the improvement of thrombus formation and/or compliance mismatch [19,20,21] are expected by using softer SF grafts.

In this paper, we developed a softer biodegradable SF graft by coating a biodegradable polyurethane (PU)-SF composite sponge on a double-raschel knitted SF vascular graft with a small diameter. In our previous paper [22], we synthesized biodegradable PU based on polycaprolactone (PCL) and prepared the regenerated biodegradable PU-SF composite fibers. PCL is a nontoxic and synthetic aliphatic polyester approved by the Food and Drug Administration and has biodegradable and biocompatible properties with a very low glass transition temperature [23,24,25,26,27]. Therefore, the biodegradable PU-SF composite material is considered a starting material for the coating of the SF graft. Then, an improvement was made by preparing water-dispersed biodegradable PU. During the process of the improvement, three kinds of ^13^C solid-state nuclear magnetic resonance (NMR) methods, namely carbon-13 (13C) cross-polarization/magic angle spinning (^13^C CP/MAS), ^13^C dipolar decoupled MAS (^13^C DD/MAS), and ^13^C refocused insensitive nuclei enhanced by polarization transfer (^13^C r-INEPT) NMR, were used for the characterization of the PU-SF coating material in the dry and hydrated states [28,29,30,31,32,33]. Then, the water-dispersed biodegradable PU-SF composite sponges were implanted in the dorsal subcutaneous tissue of rats to examine their biodegradation in vivo. Finally, 1.5 mm-diameter SF vascular grafts coated with the PU-SF composite sponge or with PU sponge only (control) were prepared and implanted in rats to investigate the patency rate and remodeling ability of the SF graft.

## 2. Materials and Methods

### 2.1. Materials

Isophorone diisocyanate (IPDI) was purchased from Evonik Japan Co., Ltd. (Tokyo, Japan); hexamethylene diisocyanate (HDI), from Tosoh Co., Ltd. (Tokyo, Japan); and polycaprolactone diol (PCL diol MW: 2000), from Daicel Corporation (Osaka, Japan). 2,2-*bis* (Hydroxymethyl)propionic acid (DMPA), 2-propanol (IPA), 3-methyl-1,5-pentandiol (MPD), polyethylene glycol monomethyl ether (MW: 400), 1,4-butanediamine (BDA), butylamine (BA), *N*,*N*-dimethylformamide (DMF), methylethylketone (MEK), and triethylamine (TEA) were purchased from Tokyo Chemical Industry Co., Ltd. (Tokyo, Japan). Zirconium octylate was obtained from Daiichi Kigenso Kagaku-Kogyo Co., Ltd. (Fukui, Japan).

### 2.2. Preparation of Biodegradable PU and Water-Dispersed Biodegradable PU Samples

Preparation of biodegradable PU was reported previously [22]. Namely, the HDI-IPA allophanate and HDI-MPD allophanate derivatives were synthesized in two and three steps, respectively. The preparations of biodegradable PU and water-dispersed biodegradable PU are the same until this process, but the next process is different. That is, for biodegradable PU, PCL diol (20.5 g), HDI-IPA allophanate (6.6 g), IPDI (2.7 g), and DMPA (0.95 g) were reacted in MEK (32.5 g) for 5 h at 75 °C. Next, the HDI-MPD allophanate derivative (3.1 g) was added to the reaction product and stirred for 1 h at 40 °C. BDA (0.82 g) in DMF (32.5 g) was added and stirred for 1 h. Then, the remaining NCO group was quenched with BA (0.34 g). Finally, a biodegradable PU solution was obtained.

On the other hand, for water-dispersed biodegradable PU, PCL diol (259.2 g), HDI-IPA allophanate (42.1 g), IPDI (17.6 g), and DMPA (6.1 g) were reacted in MEK (341.3 g) for 5 h at 75 °C. Next, the HDI-MPD allophanate derivative (16.2 g) was added to the reaction product and stirred for 1 h at 40 °C. BDA (5.7 g) was added and stirred for 1 h. Then, the remaining NCO group was quenched with BA (1.2 g). The PU solution was cooled to 25 °C, and TEA (4.1 g) was mixed at 25 °C for 0.5 h. Next, a NaOH aqueous solution (825.0 g, 0.22 *w*/*w*%) was added into the PU solution while churning up the PU solution. After churning for 1 h, MEK (442.5 g) and water (173.2 g) were evaporated from the dispersion. Finally, a water-dispersed biodegradable PU was obtained. In the ^13^C solid-state NMR spectrum of this biodegradable PU sample reported previously [22], the peaks from structural units of PCL, HDI-IPA allophanate, and HDI-MPD allophanate derivative shown in Figure 1 could be detected. On the other hand, the other compounds, IPDI, DMPA, BDA, and BA, could not be detected because of the very small contents in the PU sample.

### 2.3. Characterization of Water-Dispersed Biodegradable PU Sample

The viscosity, average particle size, solid content, and molecular weight of water-dispersed biodegradable PU samples were determined. Namely, the viscosity was measured by a single cylindrical rotational viscometer, Bismetron (Shibaura System Co., Ltd., Tokyo, Japan), under conditions of 50% relative humidity at 25 °C. The average particle size was measured by a light scattering photometer, ELSZ-2000 (Otsuka Electronics Co., Ltd., Osaka, Japan) under conditions of 50% relative humidity at 25 °C, and the data were analyzed by cumulant method. The concentration of the aqueous solution was 1.7 wt%. The solid content was obtained from the difference in weight before and after drying. The molecular weight was determined by gel permeation chromatography, HLC-8220 (Tosoh Co., Ltd., Tokyo, Japan) using column TSK gel (Tosoh Co., Ltd., Tokyo Japan). The interpretation of the results was based on the conventional calibration of columns with polystyrene standards.

### 2.4. Preparation of a Biodegradable PU-SF Film (Sample I), Biodegradable PU-SF Precipitate (Sample II), and Water-Dispersed Biodegradable PU-SF Sponge (Sample III)

The SF aqueous solution was prepared as follows [29,30,31,32,33]: SF threads wound from cocoons were degummed in an aqueous solution containing a mixture of sodium carbonate (8%, *w*/*v*) and Marseille soap (12%, *w*/*v*) at 95 °C for 120 min to remove the silk sericin (SS) completely, followed by rinsing with distilled water several times. Then, the degummed SF fibers were dissolved in a CaCl_2_-H_2_O-EtOH solution (molar ratio: 1:8:2) at 10% (*w*/*v*) for 1 h at 70 °C. This solution was filtrated to remove residual solid components and then dialyzed with a cellulose dialysis membrane tube (36/32, MWCO14,000, Viskase Companies, Inc., Willowbrook, IL, USA) against distilled water at 4 °C. The water was changed twice a day for 3 days to obtain an SF aqueous solution (concentration: 3.5 *w*/*w*%).

The biodegradable PU and SF (1:1 *w*/*w*) were dissolved in 1,1,1,3,3,3-hexafluoro-2-propanol (HFIP) and then dried (Sample I). Next, the HFIP solution of the PU-SF (1:1 in *w*/*w*) was poured into water, thereby obtaining a precipitate (Sample II). On the other hand, the SF aqueous solution was mixed with glycerin (Glyc; 1:1 in *w*/*w*) [30,32], and then the SF-Glyc mixed aqueous solution was mixed again with water-dispersed biodegradable PU (SF/PU ratio: 1:1 in *w*/*w*). The aqueous solution of the SF-PU-Glyc mixture was frozen at −20 °C overnight. Then, the frozen sponge was thawed in distilled water and bagged after Glyc removal. The SF-PU sponge pouch with water was sterilized in an autoclave at 120 °C for 20 min (Sample III). These three kinds of samples were used for the NMR observations.

### 2.5. ^13^C Solid-State NMR Observations of the PU-SF Composite Materials in Dry and Hydrated States

All ^13^C solid-state NMR spectra of the three kinds of SF-PU samples were recorded using a Bruker Avance 400 NMR spectrometer with a 4 mm double-resonance MAS probe and MAS rate of 8.5 kHz at room temperature. During the ^13^C solid-state NMR observations in the hydrated state, Samples II and III were inserted carefully in a zirconia rotor, and a PIFE cap was sealed with a PTFE insert to prevent dehydration of the hydrated samples [28,29,30,31,32,33]. The typical experimental parameters for the ^13^C CP/MAS NMR experiments included a 3.6 μs ^1^H 90° pulse, 2 ms ramped CP pulse with 71.4 kHz rf field strength, TPPM ^1^H decoupling during acquisition, 2176 data points, 8 k scans, and 3 s recycle delay. Details of the NMR conditions for the DD/MAS NMR experiments were described in previous papers [34,35]. A recycle delay of 5 s and a ^13^C 90° pulse of 3.6 μs was used. The typical experimental parameters for the r-INEPT NMR experiments entailed 3.6 μs ^1^H and 3.6 μs ^13^C pulses, an inter-pulse delay of 1/4 ^1^J_CH_ (^1^J_CH_ = 145 Hz), a refocusing delay of 1/3 ^1^J_CH_, TPPM ^1^H decoupling during acquisition, 1438 data points, 8 k scans, and 5 s recycle delay. The ^13^C chemical shifts were calibrated externally through the methylene peak of adamantane observed at 28.8 ppm relative to tetramethylsilane at 0 ppm.

### 2.6. In Vivo Degradation Test of a Water-Dispersed Biodegradable PU-SF Sponge (Sample III)

Cylindrical water-dispersed biodegradable PU-SF sponges (2.5 mm thickness × 5 mm diameter) were implanted in the dorsal subcutaneous tissue of 10 rats. They were removed at 2-week intervals until 24 weeks after implantation. For the in vivo degradation test of the PU-SF sponges, rats were anesthetized using an intraperitoneal injection of pentobarbital (50 mg/kg of body weight). The dorsal skin was shaved and disinfected, and then an incision was made with a scalpel. After sliding the PU-SF sponges into the incised surgical site, the skin was closed with 6-0 monofilament nylon sutures (Bear Medic Corporation, Tokyo, Japan). After reaching the planned implant period, the PU-SF sponges were carefully removed together with the surrounding tissue and skin. The area of the PU-SF sponges was measured and calculated using the All-in-One fluorescence microscope (Keyence BZ-9000, Keyence, Osaka, Japan).

### 2.7. Preparation of SF Vascular Grafts Coated with a Water-Dispersed SF-PU Composite Sponge (PU-SF-Coated SF Grafts) and Water-Dispersed PU Sponge Only (PU-Coated SF Grafts)

An SF-Glyc mixed aqueous solution (1:1 in *w*/*w*) was mixed with a water-dispersed biodegradable PU aqueous solution (SF/PU ratio: 1:1 in *w*/*w*). The preparation of the small-diameter SF vascular graft coated with the PU-SF sponge is summarized in Figure 2. The SF double-raschel knitted tube (base of the graft) of the SF threads with reduced amounts of SS was prepared using a computer-controlled double-raschel knitting machine (Fukui Wrap Knitting Co. Ltd., Fukui, Japan) [36,37]. Here, it was necessary to use such SF fibers with a small amount of SS to knit on the knit machine. ① The knitted silk tube was then degummed to remove the remaining SS completely. ② A PTFE rod with a 1.5 mm diameter was inserted in the SF tube with a 1.5 mm diameter. ③ The rod covered with the SF tube was immersed into a pipe filled with the mixed aqueous solutions of SF-PU-Glyc for coating. The pipe was placed in a desiccator, and then the desiccator interior was kept under a reduced pressure of 100 hPa until no air bubbles appeared from the coated surface of the SF tube. ④ After the treatment, the SF graft was frozen at −20 °C overnight. ⑤ and ⑥ Then, the graft was immersed in distilled water and bagged with distilled water. ⑦ The pouch SF graft in distilled water was sterilized in an autoclave at 120 °C for 20 min. For the preparation of SF vascular grafts coated with a water-dispersed PU sponge only, the preparation process was slightly different from the SF-PU-coated SF graft. That is, stages ① and ② are the same. Then, the rod covered with the SF tube was immersed into a pipe filled with the water-dispersed PU solution for coating in stage ③. Then, the PU-coated SF graft was freeze-dried after freezing the graft in liquid nitrogen. Finally, the graft was bagged with distilled water, and the pouch SF graft in distilled water was sterilized in an autoclave at 120 °C for 20 min [17,32]. The scanning electron microscopy (SEM; VE-7800 Keyence, Japan) was used to observe both the PU-SF composite sponge and the SF double-raschel knitted graft coated with the PU-SF composite sponge before implantation.

### 2.8. Mechanical Properties of the SF Double-Raschel Knitted Tube Coated with a Water-Dispersed PU-SF Composite Sponge and SF Double-Raschel Knitted Tube without Coating

Because the diameter of 1.5 mm of the SF double-raschel knitted tube is too small for tensile testing, we used an SF double-raschel knitted tube with a 3.5 mm diameter prepared by the same knitting matter as that with 1.5 mm diameter. The tensile testing of the SF double-raschel knitted tube coated with PU-SF composite sponge and SF double-raschel knitted tube without coating was performed in the hydrated state using an EZ-Graph tensile testing machine (EZ-Graph, Shimadzu Co. Ltd., Kyoto, Japan) at room temperature according to the methods reported previously [32,36]. Namely, the samples were immersed in distilled water for 24 hrs, and then the stress-strain curves were observed immediately after wiping the water on the surface of the samples. The breaking strength (N) and elongation at break (%) were calculated from the stress-strain curves (*n* = 6). The thickness and the standard deviation (*n* = 10 each) of the hydrated SF double-raschel knitted tubes without and with PU-SF composite sponge coating were determined using a micrometer (MDC-25PJ, Mitutoyo Co., Kawasaki, Japan).

### 2.9. Implantation of the SF Vascular Grafts in Rats

Two types of SF grafts (3 cm in length × 1.5 mm in inner diameter) coated with PU-SF or only PU sponges were implanted in the abdominal aorta of 24 rats under a stereoscopic microscope (LEICA M60; Leica Microsystems, Tokyo, Japan). Half of the rats were sacrificed at 2 weeks after implantation, and the remaining rats were sacrificed at 4 weeks after implantation. For graft implantation, the rats were anesthetized using an intraperitoneal injection of pentobarbital (50 mg/kg of body weight). The abdominal aorta was carefully exposed, and the aortic branches in this segment were ligated. After an intravenous injection of heparin (100 IU/kg), the proximal and distal portions of the infrarenal aorta were clamped with non-crushing vascular clamps. A 1 cm length of the segment of the aorta was removed and replaced with grafts by end-to-end anastomosis using interrupted 9–0 monofilament nylon sutures (Bear Medic Corporation, Tokyo, Japan), starting with two stay sutures at 180° to each other and then suturing the front wall followed by the back wall. Each anastomosis required 8–10 stitches. The distal and then proximal vascular clamps were removed slowly, and flow was restored through the grafts. Graft patency was visually confirmed. No anticoagulant or antiplatelet agent was administered postoperatively. Prior to euthanasia, the rats were perfused with a 0.9% saline solution through the left ventricle. The grafts were carefully removed together with the surrounding tissues.

### 2.10. Histopathological Examination

The implanted SF graft was hollowed out together with the skin with surgical scissors and divided in half. In addition, the central part of the graft was cut 4 mm transversely, and the sutured part of the remaining native blood vessel and artificial vascular graft was cut longitudinally. These samples were fixed with ethanol for histological analyses. The fixed samples were embedded in paraffin and processed for hematoxylin and eosin (H&E) and Masson’s trichrome (MTC) staining. Sections for immunohistochemical staining were incubated with primary antibodies, including α-smooth muscle actin (α-SMA; clone 1A4; Sigma-Aldrich, Inc., Tokyo, Japan) and CD31 anti-rat antibody (BD Biosciences, Inc., Tokyo, Japan). The A-SMA sample was incubated with biotinylated anti-mouse immunoglobulin secondary antibody (Vector Laboratories, Inc., Burlingame, CA, USA), and subsequent color development was obtained using Vectastain ABC-AP (Vector Laboratories, Inc., Burlingame, CA, USA). The CD31 sample was incubated with N-Histofine Simple Stain Rat MAX PO (Nichirei Biosciences, Inc., Tokyo, Japan), and subsequent color development was obtained using Vectastain ABC-AP (Vector Laboratories, Inc., Burlingame, CA, USA).

### 2.11. Animals

Sprague-Dawley rats (Charles Laboratories, Yokohama, Japan) weighing 200–300 g were used for in vivo study. All the rats were kept in micro-isolator cages with a 12 h light/dark cycle. All experimental procedures and protocols were approved by the Tokyo University of Agriculture and Technology (TUAT; approval No. R02–25). The rats were managed and cared for in accordance with the standards established by the TUAT and described in its “Guide for the Care and Use of Laboratory Animals.”

## 3. Results and Discussion

### 3.1. Examination of Biodegradable PU-SF Composite Materials for Coating Knitted SF Grafts

In this paper, three kinds of biodegradable PU-SF composite materials (samples I, II, and III) were used to characterize the structure and dynamics of PU and SF in the composites using ^13^C solid-state NMR. The biodegradable PU and SF (1:1 in *w*/*w*) were dissolved in HFIP and then dried (Sample I). The HFIP solution of the PU-SF (1:1 in *w*/*w*) was poured into water, and a precipitate was obtained (Sample II). The aqueous solution of the SF-PU-Glyc mixture was frozen at −20 °C overnight. The frozen sponge was thawed in distilled water and bagged after Glyc removal. The SF-PU sponge pouch with water was sterilized in an autoclave at 120 °C for 20 min (Sample III).

#### 3.1.1. ^13^C CP/MAS NMR Spectra of Samples I and II in Dry States

In our previous paper [22], we synthesized biodegradable PU based on PCL; therefore, the biodegradable PU-SF composite material is considered a starting material. The ^13^C CP/MAS NMR spectrum of Sample I in the dry state is shown in Figure 3a. The peaks from PU and SF could be observed and assigned as shown in the figure, in accordance with previous reports [28,29,30,31,32,33,34,35]. The asymmetrical peaks at 88 ppm were the spinning sideband (ssb) of the carbonyl peaks. The peaks from 1 to 6 of PCL unit, the peak 7 of HDI-IPA allophanate, and the peak 8 of the HDI-MPD allophanate derivative (Figure 1) could be observed from PU. The ^13^C chemical shifts of the PU peaks in Sample I were essentially the same as those of pure PU reported previously [22]. On the other hand, the main peaks of the Ala Cβ, Gly Cα, Ala Cα, and Ser Cα carbons in SF were observed at 15.9, 44.5, 52.0, and 59.6 ppm, respectively. The ^13^C chemical shift values mean that SF in Sample I took 3_10_ helix conformations. This conformation is the same as that of pure SF film dried from the HFIP solution, as reported previously [38,39]. Thus, the conformation of SF showed no changes even if PU coexisted in Sample I.

Next, the ^13^C CP/MAS NMR spectrum of Sample II in the dry state is shown in Figure 3b. The ^13^C chemical shifts of the peaks from PU did not change even in this process. However, the chemical shifts of the peaks of SF changed significantly. That is, the conformation of SF was a mixture of β-sheet (β*) and random coil (rc) judging from the conformation-dependent ^13^C chemical shift, as will be described in detail.

#### 3.1.2. ^13^C Solid-State NMR Spectra of Sample II in Hydrated States

The combination of three kinds of ^13^C solid-state NMR methods, namely ^13^C r-INEPT, ^13^C DD/MAS, and ^13^C CP/MAS, has provided useful information on the structure and dynamics of SF in the hydrated state [29,30,31,32,33]. The ^13^C r-INEPT where the pulse sequence was developed for solution NMR selectively observes the mobile components of the hydrated SF chains with fast isotropic motion (>10^5^ Hz) [40]. By contrast, ^13^C CP/MAS NMR selectively observes the immobile components of SF chains or those with very slow motion (<10^4^ Hz) [40]. If the penetration of water molecules causes an increase in the local chain mobility of heterogeneous SF, a loss of CP signal of the amino acid residues occurs. Consequently, such a mobile domain cannot be observed in the ^13^C CP/MAS NMR spectra [35,41]. In addition, ^13^C DD /MAS NMR can be used to obtain structural information on both the mobile and immobile domains in the SF chains in the hydrated state. Therefore, these three kinds of ^13^C solid-state NMR methods were also used for the characterization of PU-SF composite materials.

Figure 4 shows the expanded regions (10–80 ppm) in the (a) ^13^C r-INEPT, (b) ^13^C DD/MAS, and (c) ^13^C CP/MAS NMR spectra of Sample II in the hydrated state together with the assignments. The ^13^C CP/MAS NMR spectrum was essentially the same between the dry and hydrated states (Figure 3b and Figure 4c). It is noted that the three spectra (Figure 4a–c) in the hydrated state are quite different. Thus, there are quite different components from the viewpoint of dynamics in these PU-SF composite materials, and the samples are quite heterogeneous in the hydrated state. Especially the peaks from the PU became very small or disappeared in the ^13^C r-INEPT spectrum. In addition, we noticed that the chemical shifts of peaks 1 and 5 were slightly different between (b) the ^13^C DD/MAS and (c) ^13^C CP/MAS NMR spectra. The double-split peaks for the C=O and two methylene carbons (1 and 5) were observed in the gated decoupled/MAS ^13^C NMR spectrum of PCL and assigned to the crystalline (c) and non-crystalline (n) regions of PCL by Kaji and Horii [42]. In addition, Schäler et al. [43] supported their assignment and reported an additional double-split peak for the methylene carbon, 2. The difference in the ^13^C chemical shift between the (c) and (n) peaks can be explained in terms of the ^13^C γ-gauche effect of the alkyl chain [43,44]. Thus, only the peaks from the crystalline regions of PU (c) could be observed in the ^13^C CP/MAS spectrum, and those of the non-crystalline regions (n) could be additionally observed in the ^13^C DD/MAS spectrum. As the peaks from the crystalline regions were still observed, relatively broad and asymmetrical peaks appeared. The significant changes were that peaks 2, 3, 4, and 5 from the PCL of PU became very small, and peak 1 disappeared in the ^13^C r-INEPT spectrum. This is interpreted as follows: The hydrophobic alkyl chains of PCL aggregated with each other by van der Waals forces and formed the inner core in water. By contrast, remarkable increases were observed for the peaks at 69.7 and 58.2 ppm in the ^13^C r-INEPT spectrum, which were assigned to the methylene carbon 8 and terminal methyl carbon 9, respectively, in the repeated hydrophilic EG units with Mn of 400 in the HDI-MPD allophanate derivative (Figure 1). Thus, the repeated hydrophilic EG units move freely in water together with the Cα and Cβ carbons of the hydrated amino acid residues with the random coil conformation of SF. The ^13^C chemical shifts, together with the peak assignments, are summarized in Table 1.

#### 3.1.3. ^13^C Solid-State NMR Spectra of Sample III in Hydrated States

Next, we prepared Sample III by using water-dispersed PU described in Section 2.1.

The viscosity, average particle size, solid content, and molecular weight of the water-dispersed PU sample are as follows. The viscosity; 125 mPa^.^s, average particle size; 180 nm, solid content; 35% and number averaged molecular weight (Mn); 19200.

As SF is soluble in water, the water-dispersed PU can mix with SF easily in an aqueous solution. In addition, Glyc was used as the porogen to prepare a water-dispersed PU-SF composite sponge because we used Glyc frequently for the preparation of SF sponge for SF grafts [32,45,46]. Figure 5 shows the expanded regions (10–80 ppm) in the (a) ^13^C r-INEPT, (b) ^13^C DD/MAS, and (c)^13^C CP/MAS NMR spectra of Sample III in the hydrated state together with the assignments. The ^13^C chemical shifts, together with the peak assignments, are summarized in Table 2. The ^13^C CP/MAS NMR spectrum (III) of the water-dispersed PU-SF sponge is quite similar to that of Sample II, as shown in Figure 4. Similarly, both ^13^C DD/MAS NMR spectra (Figure 4b and Figure 5b) were similar to each other. However, the ^13^C r-INEPT spectra were quite different, especially for the peaks from PU. That is, contrary to Figure 4a, the alkyl peaks from 1 to 5 of the PCL unit in Sample III created sharp and strong peaks. Kaji and Horii [42] also determined the ^13^C spin-lattice relaxation times, the T_1_ value of each carbon of PCL. Three T_1_ values of all methylene carbons were 232–279, 25–30, and 0.19–0.26 s. The former two components (both crystalline components) had relatively longer T_1_ values than the latter component (non-crystalline component). Thus, the part of the alkyl chains of PCL in the non-crystalline component is in the mobile state with the shortest T_1_ values and did not form in the aggregated state. This difference in the ^13^C r-INEPT spectra seems to originate from the difference in the process of the sample preparations. Thus, SF and PU were more randomly mixed with each other in Sample III and more flexible in water. They seem more suitable water-dispersed PU-SF composite sponges for coating knitted SF grafts.

### 3.2. Implantation Experiments with Rats

#### 3.2.1. Water-Dispersed Biodegradable PU-SF Composite Sponge

The PU-SF composite sponge before implantation is shown in Figure 6a. In addition, the SEM picture shown in Figure 7a indicates that there are 50–100 μm holes in the inner part of the sponge. During the PU-SF sponge implantation in the rats, no inflammatory reaction was observed at the implanted site at all times. At 2 weeks after implantation, the implanted site was swelled by the sponge and could be seen directly. The implanted subcutaneous sponge was covered with a thin membrane, and angiogenesis was observed immediately above it (Figure 6b,c). At 8 weeks after implantation, the swelling of the skin caused by the sponge became invisible to the naked eye and could only be confirmed from the subcutaneous view (Figure 6d,e). At 6 months after implantation, the sponge did not completely disappear but became a thin film-like structure (Figure 6f,g). The area of the sponge decreased to less than half at 4 weeks after implantation and then gradually decreased as the implantation period increased. The sponge area before implantation was 12.5 mm^2^, the sponge area 8 weeks after implantation was 2.626 mm^2^, and the sponge area 6 months after implantation was 0.4623 mm^2^. During any period, the implanted sponge did not cause adverse reactions such as erythema of the skin, fistula, and granulomatous formation, and the volume gradually decreased as the implantation period increased. As inflammatory cells infiltrated and decomposed the PU-SF sponge at 2 weeks after implantation, even if the PU was mixed with SF, it showed biodegradability without impairing the properties of the conventional SF. When porous SF is implanted into soft tissue, it is decomposed 1 to 3 weeks after implantation, and infiltration of blood vessels and autologous tissue begins [12]. Therefore, the PU-SF sponge produced in this study is considered to have biodegradability and biocompatibility as an in vivo material.

#### 3.2.2. Tensile Testing of the SF Double-Raschel Knitted Graft without and with PU-SF Composite Sponge

As mentioned in Materials and Methods, we used an SF double-raschel knitted tube with a 3.5 mm diameter for tensile testing. The tensile testing experiment was performed for the SF double-raschel knitted tube without coating and the SF double-raschel knitted tube with PU-SF composite sponge in the hydrated state. All of the data of the breaking strength (N) and elongation-at break (%) were listed in Table 3, and the stress-strain curves were obtained as shown in Figure 8a,b. The averaged values of the breaking strength (N) and elongation at break (%) were calculated to be (a) 29.1 ± 1.1 and 147 ± 11, and (b) 35.5 ± 3.8 and 120 ± 9, respectively. Namely, the breaking strength increases slightly, and elongation at break decreased slightly by SF-PU sponge coating, but the difference in the mechanical properties is relatively small. This is due to small amounts of coating PU-SF sponge covered the SF tubes. Actually, the values of thickness for SF knitted tubes without and with coating were 0.693 ± 0.025 mm and 0.693 ± 0.029 mm, respectively, which means exactly the same value with each other. In addition, as shown in Figure 7b, the space among woven silk fibers of the SF grafts was only lightly covered with the PU-SF composite sponge.

#### 3.2.3. SF Grafts Coated with Only the PU and Water-Dispersed Biodegradable PU-SF Composite Sponges

The appearance of two types of SF grafts immediately 2 and 4 weeks after implantation are shown in Figure 9. In graft implantation, two types of SF grafts coated with a PU sponge (Figure 9a) and a PU-SF sponge (Figure 9e) were prepared. As mentioned above, the space among woven silk fibers of the grafts was only lightly covered with the PU-SF sponge (Figure 7b). All 24 implantations were successfully performed without complications (e.g., uncontrolled bleeding). After refluxing the blood, bleeding from the PU-coated graft was not observed, with only slight leakage of blood from the anastomotic site (Figure 9b). On the other hand, mild blood leakage was observed from both the PU-SF-coated SF graft and anastomotic site (Figure 9f). The bleeding was stopped by compression with a cotton swab for a few seconds. No significant differences were observed among the grafts with respect to the ease of handling and resistance to suture. All the rats survived until the scheduled date of removal and were sacrificed at 2 or 4 weeks after implantation. After 2 and 4 weeks, occlusion was observed in 5 of 6 rats with PU-coated SF grafts. On the other hand, patency was observed in all the rats with PU-SF-coated SF grafts. All the grafts were covered with a thin layer of adipose and surrounding tissue (Figure 9c,d,g,h). In addition, peeling of the surrounding tissue from the grafts was easy, with slight bleeding observed from the peeling sites. Although the physical properties of the SF graft were not tested in this study, no side reactions were observed during and after implantation, which suggests that the strength was sufficient for implantation. The strength of the SF grafts is affected by the base structure and coating [32], but because the base and coating created in this study were based on those created in a previous study, the physical properties of the SF graft were considered not to differ much from those of a previously reported SF graft [45]. However, if the animals to receive implantation differ, the required strength will also be different, so reevaluation will be essential when implanting in larger animals in the future.

### 3.3. Histopathological Examination

#### 3.3.1. Water-Dispersed Biodegradable PU-SF Composite Sponge

The histological images of H&E staining 2 weeks and 6 months after implantation are shown in Figure 10. The PU-SF sponge was covered with a thin film-like structure at 2 weeks after implantation (Figure 10a,b). Various inflammatory cells such as neutrophils, lymphocytes, macrophages, and multinucleated giant cells gathered around the SF, and the PU-SF sponge was fragmented. At 6 months after implantation (Figure 10c,d), the thickness of the PU-SF sponge almost disappeared and was further subdivided. Many biomaterials using SF-PU composite materials have been actively studied in the cardiovascular field [20,47,48,49,50,51,52,53,54]. However, because the PU used for the composite with SF is generally not biodegradable, it was difficult to use for coating materials in combination with biodegradable SF. In small-diameter artificial SF vascular grafts, if the coated material is not decomposed even after implantation, remodeling is delayed, which causes occlusion due to thrombus formation and intimal hyperplasia [46]. Therefore, the SF sponge prepared by mixing with water-dispersed degradable PU in this study was decomposed early after implantation and was shown to be useful as a coating material for small-diameter artificial vascular grafts.

#### 3.3.2. SF Grafts Coated with Water-Dispersed Biodegradable PU and PU-SF Composite Sponges Only

The photographs of the microscopic findings and histological cross-section images of the PU- and PU-SF-coated SF grafts at 2 and 4 weeks after implantation are shown in Figure 11. The lumen diameter of the patented SF grafts did not change in any of the grafts, but narrowing was observed in the PU-coated SF graft at 4 weeks after implantation (Figure 11a,f,k,p). The H&E-stained images of the PU-SF-coated SF grafts at 2 weeks after implantation showed inflammatory cells, lymphocytes, macrophages, and neutrophils around and inside the grafts (Figure 11l). Those reactions were prominent on the outside of the grafts. On the other hand, in the PU-coated SF grafts, inflammatory cells were gathered on the outside like in the PU-SF-coated SF grafts but could not be confirmed on the inside of the grafts (Figure 11b). The MTC-stained images showed collagen fibers gathered throughout the PU-SF-coated SF grafts and infiltrated the interstices of the fibers of the base of the grafts (Figure 11m). In the PU-coated SF grafts, the collagen fibers were also gathered on the outside of the graft but hardly infiltrated into the inside of the graft (Figure 11c). The α-SMA-stained images showed that smooth muscle cells gathered mainly outside the PU-SF-coated SF grafts and along the luminal surface of the graft (Figure 11n). However, in the PU-coated SF grafts, smooth muscle cells could only be confirmed outside of the grafts (Figure 11d). The CD31-stained images showed that vascular endothelial cells were not detected in the luminal surfaces of the two types of grafts (Figure 11e,o). In the PU-SF-coated SF grafts, some blood vessels infiltrating the gaps in the base of the grafts could be confirmed (Figure 11o). In the PU-SF-coated SF grafts at 4 weeks after implantation, the number of inflammatory cells gathered on the outside of the grafts decreased (Figure 11q). In addition, the layered structure formed inside the grafts was confirmed more firmly, but the lumen was firmly secured and patent. In the PU-coated grafts, a large number of inflammatory cells were gathered on the inside of the grafts (Figure 11g). Collagen fibers had infiltrated the gaps in the base of the PU-SF-coated SF grafts, as they did 2 weeks after implantation (Figure 11r). In the PU-coated SF graft, collagen fibers were observed inside the graft 4 weeks after implantation, but not in the gap of the base of the graft (Figure 11h). Smooth muscle cells were similarly gathered inside of the graft in both SF grafts (Figure 11s,i). Endothelial cells could be confirmed in a part of the lumen surface of the implanted grafts, but they were not completely covered (Figure 11t). In the PU-coated SF grafts, vascular endothelial cells could not be confirmed as in the case at 2 weeks after implantation (Figure 11j).

As the PU-coated graft did not cause blood leakage at the time of implantation or an excessive inflammatory reaction after implantation, it was considered that there was no problem in biocompatibility. However, the patency rate after implantation was low. The histopathological results of the patency PU-coated graft did not show sufficient tissue infiltration of host cells 2 weeks after implantation. In addition, luminal stenosis due to intimal hyperplasia was observed 4 weeks after implantation. From these results, it was considered that the porous state was not obtained when the graft was coated with PU only. Luminal stenosis had occurred the only patency PU-coated graft 4 weeks after implantation. In other experiments in which artificial vascular grafts were prepared using PU, differences in tissue infiltration and remodeling due to differences in the size of the pore size were observed [55]. Therefore, whether sufficient porosity is obtained when coating with urethane alone will be necessary to consider in the future.

So far, many reports have been published about small-diameter SF vascular grafts prepared with the electrospinning method [15,17,49,56,57,58,59,60,61,62,63,64,65]. Electrospinning is the formation of fibers in the micrometer-to-nanometer range by electrically charging slowly extruded solutions [66,67,68] and has several advantages. It requires low working volumes to produce large amounts of scaffold with relatively fine control over the product. It is a promising method for preparing nanofiber scaffolds that mimic the morphological properties of natural extracellular matrices. However, further studies are required to improve its mechanical strength because the grafts prepared with the electrospinning method are generally too weak. In addition, improvement of the blood compatibilities of the grafts is also required [17]. An alternative method for the direct use of silk fibers is the generation of a knitted or weaved silk structure to reinforce three-dimensional porous tissue engineering scaffolds performed in this paper. Double-raschel knitting has been used in the preparation of vascular grafts of commercially available polyester fibers [17]. In the double-raschel knitting process, the physical or mechanical characteristics of the vascular graft and the sizes can be altered using a computer-controlled double-raschel knitting machine [36,37,45,46]. The merits of double-raschel knitting are integrated in that the silk threads do not become flat even if the threads are pressurized by a guide or needle, and the silk threads exhibit appropriate elasticity. In addition, because of the numerous contact points in the fibers, adequate strength and protection from loosening at the edges during the implantation process could be attained. Friction at the contact points is also reduced, which prevents thread tears and/or thread separation during the manufacturing process. However, the grafts should be coated to prevent blood leakage during implantation [36,37,45,46]. If the coated part dissolves after implantation, it has a structure that facilitates the infiltration of host cells required for remodeling. In the PU-SF-coated SF graft prepared in this study, the coating part was degraded 2 weeks after implantation, and the host cells infiltrated the gap of the base. In addition, vascular endothelial cells were confirmed inside the graft 4 weeks after implantation, indicating that it is useful as a coating material for SF artificial vascular grafts. In the future, we would like to examine the optimum amount of PU to be mixed with SF by conducting implant PU-SF-coated SF grafts into large animals and investigate patency and remodeling ability for practical application.

## 4. Conclusions

To develop a softer SF graft by coating with water-dispersed biodegradable polyurethane (PU) based on a polycaprolactone and SF composite sponge on the knitted SF vascular graft, several in vitro and in vivo experiments were performed. The ^13^C r-INEPT NMR spectrum of water-dispersed biodegradable PU showed that both main components of the non-crystalline domain of PU and the amorphous domain of SF were highly mobile in the hydrated state. The implantation experiments in the rats showed that the PU-SF-coated SF graft was rapidly degraded in vivo, and remodeling to self-organization was promoted. These results indicate that water-dispersed PU-SF-coated SF grafts are flexible and biodegradable and may be useful as small-diameter artificial vascular grafts.

## Figures and Tables

**Figure 1 molecules-26-04649-f001:**
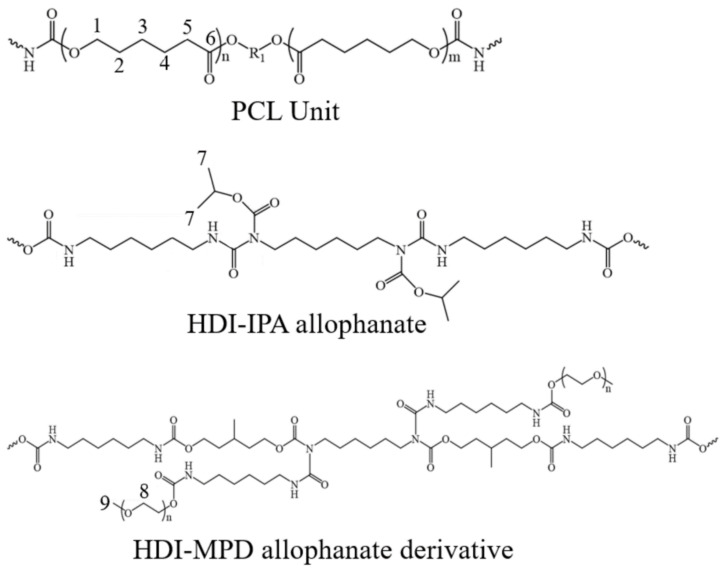
Structural units of biodegradable PU. The number of each carbon observed in the ^13^C solid-state NMR spectra in this paper is noted.

**Figure 2 molecules-26-04649-f002:**
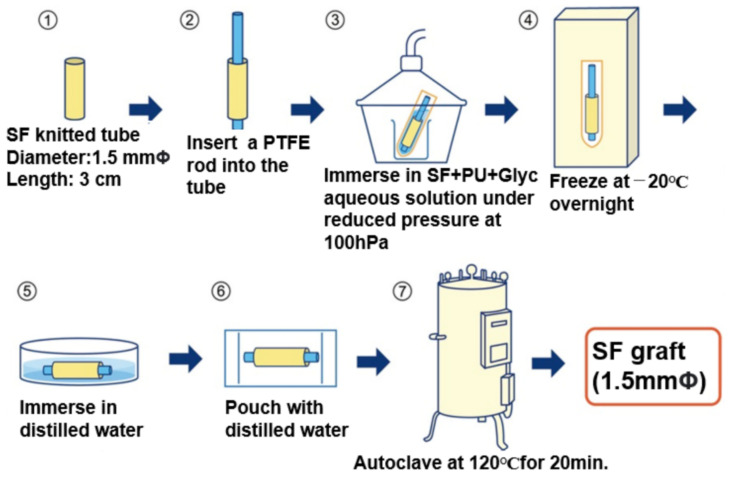
Preparation of SF vascular graft coated with a water-dispersed PU-SF composite sponge.

**Figure 3 molecules-26-04649-f003:**
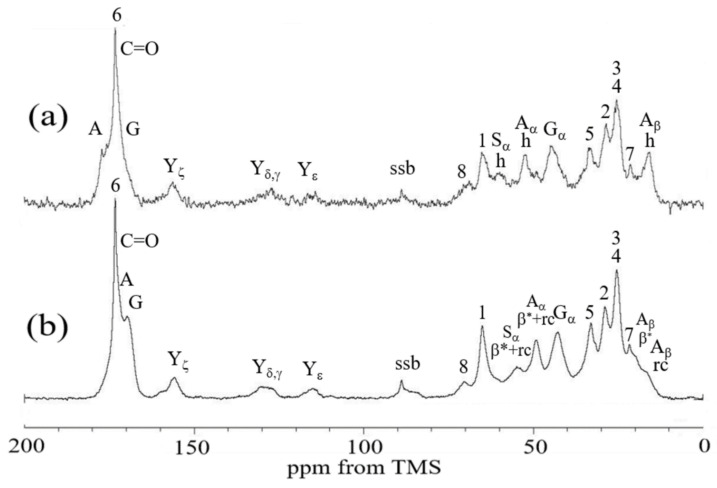
^13^C CP/MAS NMR spectra of (**a**) Sample I and (**b**) Sample II in the dry states. β*: anti-parallel β-sheet, h: 3_10_ helix, and rc: random coil. The number of each carbon indicates the PU peaks noted in Figure 1.

**Figure 4 molecules-26-04649-f004:**
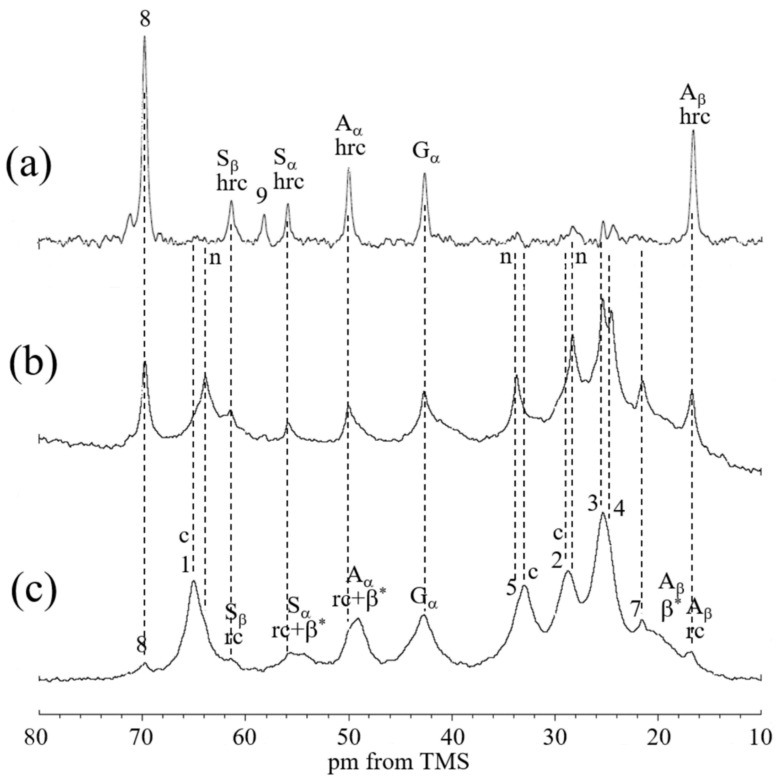
Expanded (**a**) ^13^C r-INEPT, (**b**) ^13^C DD/MAS, and (**c**) ^13^C CP/MAS NMR spectra (0–80 ppm) of Sample II in the hydrated state. β*: anti-parallel β-sheet, rc: random coil, and hrc: hydrated random coil. The number of each carbon indicates the PU peaks noted in Figure 1.

**Figure 5 molecules-26-04649-f005:**
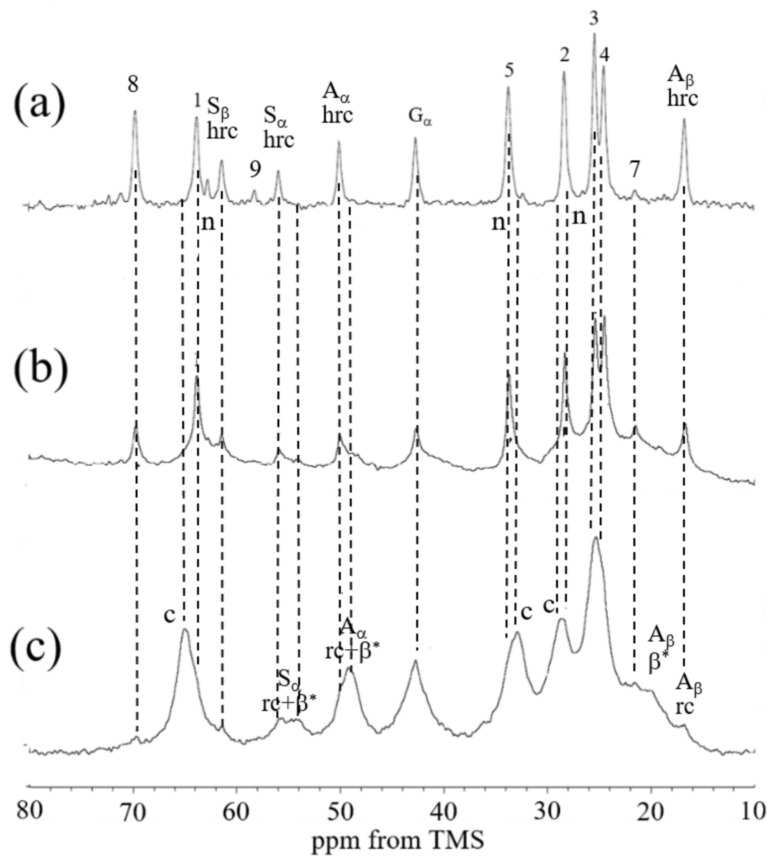
Expanded (**a**) ^13^C r-INEPT, (**b**) ^13^C DD/MAS, and (**c**) ^13^C CP/MAS NMR spectra (10–80 ppm) of Sample III in the hydrated state. β*: anti-parallel β-sheet, rc: random coil, and hrc: hydrated random coil. The number of each carbon indicates the PU peaks noted in Figure 1.

**Figure 6 molecules-26-04649-f006:**
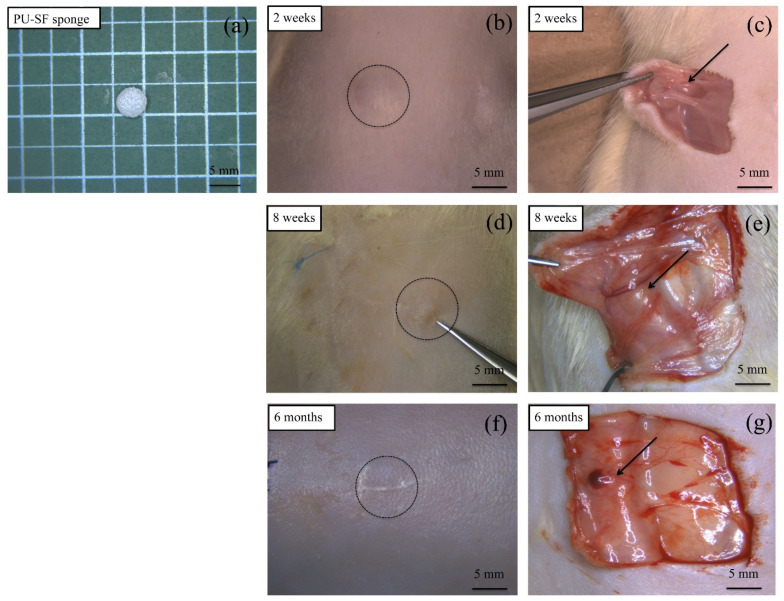
Photographs of the PU-SF sponge (**a**) before and (**b**,**d**,**f**) after implantation. Appearance image of the sponge embedding site 2 weeks after implantation (**b**) and appearance image from the inside after skin incision (**c**). Appearance image of the sponge embedding site 8 weeks after implantation (**d**) and appearance image from the inside after skin incision (**e**). Appearance image of the sponge embedding site 6 months after implantation (**f**) and appearance image from the inside after skin incision (**g**).

**Figure 7 molecules-26-04649-f007:**
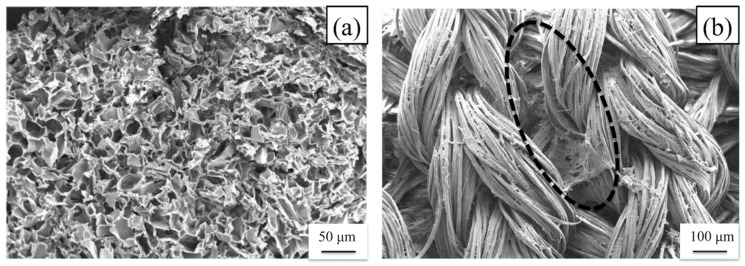
Scanning electron microscopy pictures of (**a**) PU-SF composite sponge and (**b**) outer surface of double-raschel knitted SF vascular grafts coated with PU-SF composite sponge before implantation. The part enclosed by the dotted line shows the PU-SF composite sponge.

**Figure 8 molecules-26-04649-f008:**
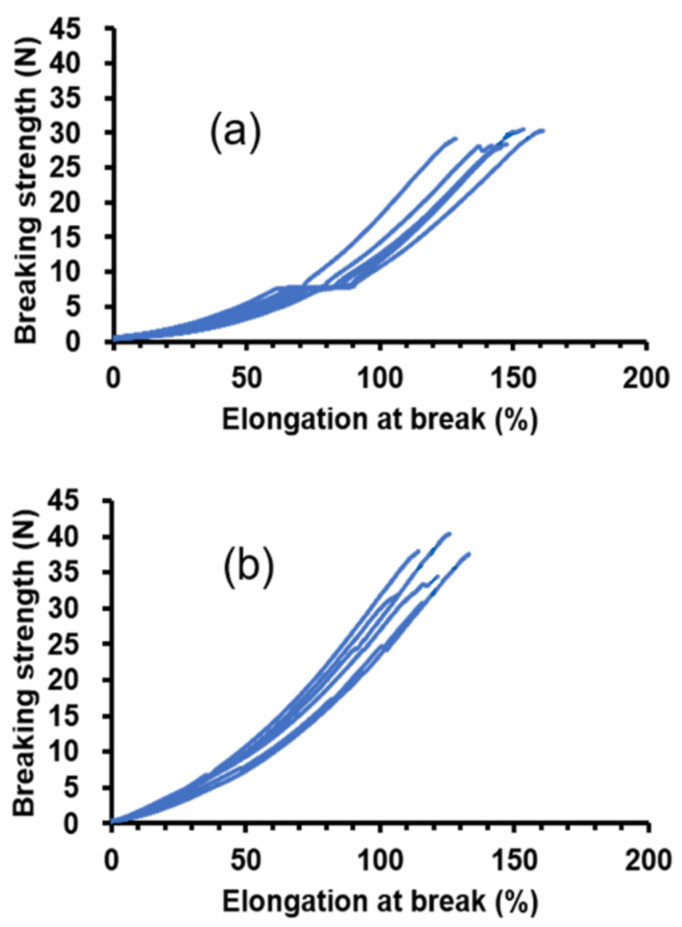
The stress-strain curves of the hydrated 3.5 mm-diameter SF knitted tubes (**a**) without and (**b**) with PU-SF composite sponge coating. The error bars are included in the figure. The averaged values of the breaking strength (N) and elongation at break (%) were (**a**) 29.1 ± 1.1 and 147 ± 11, and (**b**) 35.5 ± 3.8 and 120 ± 9, respectively.

**Figure 9 molecules-26-04649-f009:**
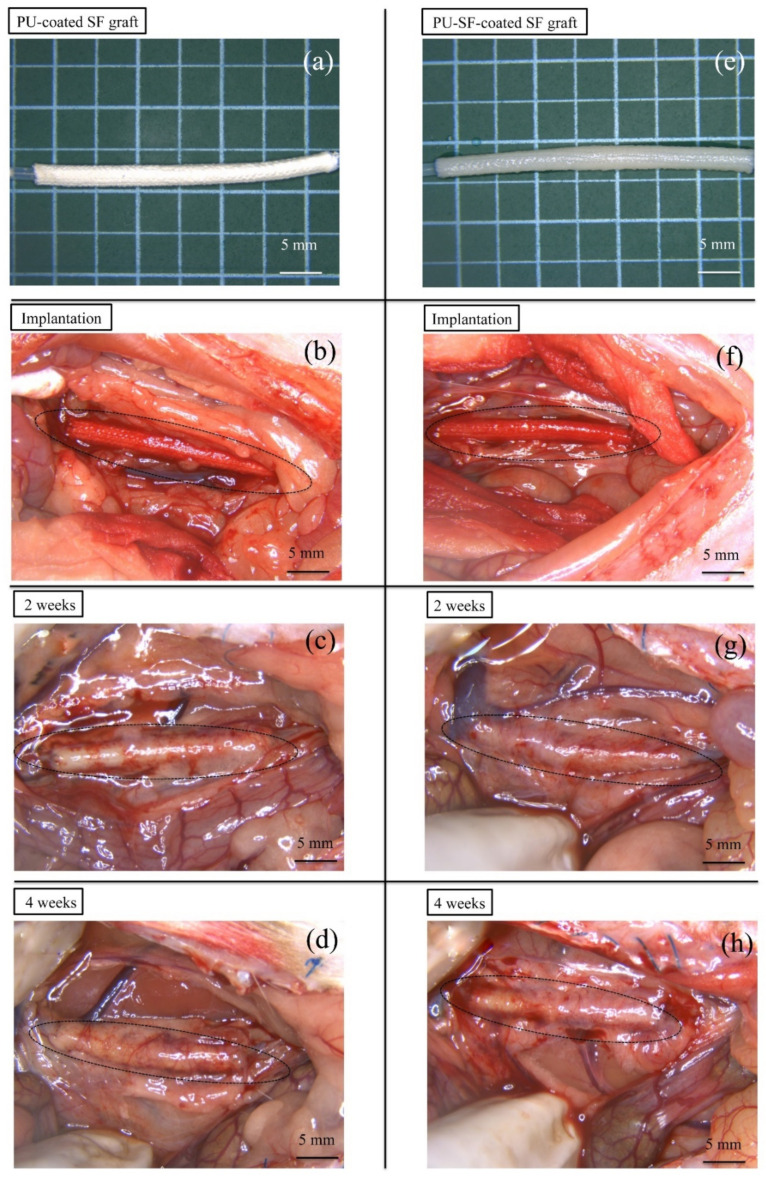
PU-coated (**a**) and PU-SF-coated SF grafts (**e**) before implantation. Photographs of the PU- (**b**) and PU-SF-coated grafts (**f**) immediately after releasing the vascular clamp after implantation. Hemostasis was achieved in all the grafts, and no deaths due to bleeding occurred. Photographs of the microscopic findings during extraction of the grafts: (**c**) the PU- and (**g**) PU-SF-coated grafts at 2 weeks after implantation, and (**d**) the PU- and (**h**) PU-SF-coated graft at 4 weeks after implantation. Bending, dehiscence, granuloma, hematoma, and aneurysm formation were not observed. Peeling of the surrounding tissue from these grafts was easy, and no fatal bleeding was observed.

**Figure 10 molecules-26-04649-f010:**
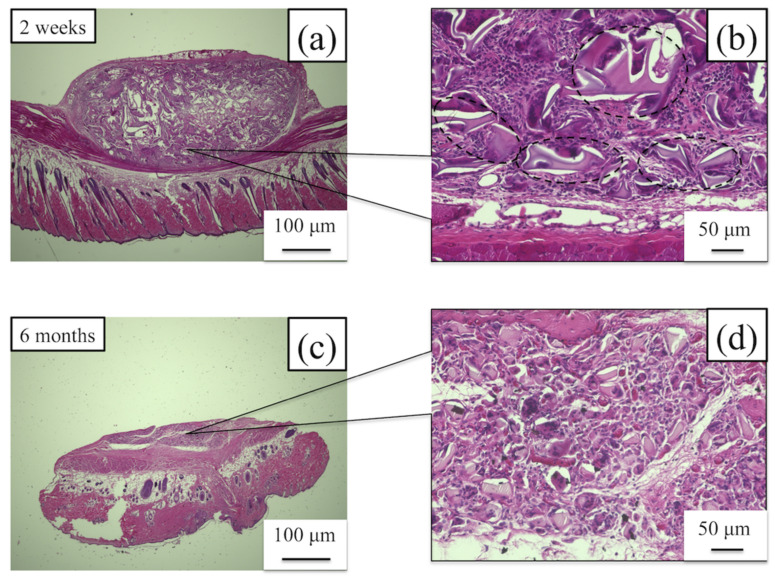
Histological images of hematoxylin and eosin staining 2 weeks after implantation of the (**a**) PU-SF sponge and (**b**) higher magnification of the PU-SF sponge. Histological images of hematoxylin and eosin staining 6 months after implantation of (**c**) the PU-SF sponge and (**d**) higher magnification of the PU-SF sponge. The part enclosed by the dotted line shows the SF sponge. At 2 weeks after implantation, inflammatory cells gathered in the gaps of the SF sponge were confirmed. In addition, the SF sponge was further decomposed 6 months after implantation.

**Figure 11 molecules-26-04649-f011:**
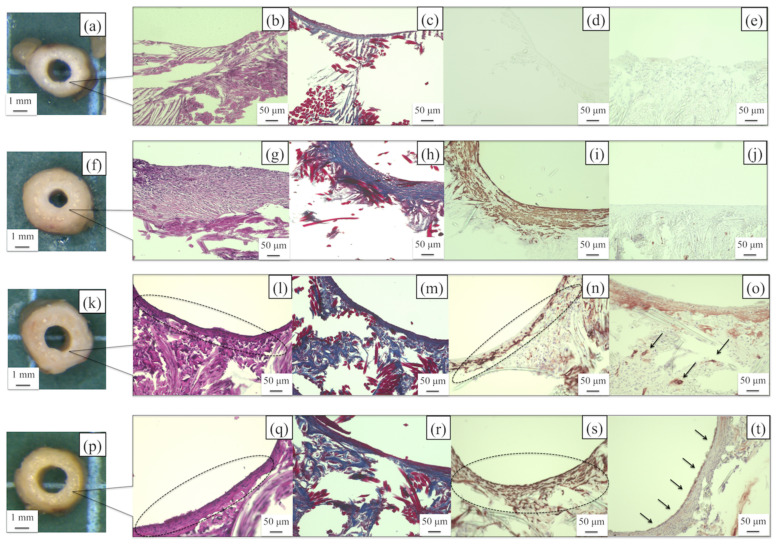
Photographs of the microscopic findings of the PU- (**a**) and PU-SF-coated SF grafts (**k**) at 2 weeks after implantation and the PU- (**f**) and PU-SF-coated grafts (**p**) at 4 weeks after implantation. Histological cross-sectional images of the PU-coated graft at 2 weeks after implantation: the PU-coated graft after (**b**) H&E, (**c**) EVG, and (**d**) α-SMA staining, and (**e**) the PU-SF-coated graft after CD31 staining. Histological cross-sectional images of the PU-coated graft at 4 weeks after implantation: the PU-coated graft after (**g**) H&E, (**h**) EVG, (**i**) α-SMA, and (**j**) CD31 staining. Histological cross-sectional images of the PU-SF-coated graft at 2 weeks after implantation: the PU-SF-coated graft after (**l**) H&E, (**m**) EVG, (**n**) α-SMA, and (**o**) CD31 staining. Histological cross-sectional images of the PU-SF-coated graft at 4 weeks after implantation: the PU-SF-coated graft after (**q**) H&E, (**r**) EVG, (**s**) α-SMA staining, and (**t**) CD31 staining. The part enclosed by the dotted line shows the intima. The arrows indicate that vascular endothelial cells adhered to the inner surfaces of the grafts.

**Table 1 molecules-26-04649-t001:** ^13^C chemical shifts (ppm) of ^13^C CP/MAS, ^13^C DD/MAS, and ^13^C r-INEPT NMR spectra of Sample II in the hydrated state together with the assignments.

Number of Carbons	CP/MAS	DD/MAS	INEPT
1	65.0 (c)	63.8 (n)	—
2	28.7 (c)	28.3 (n)	28.3 (n)
3	25.4	25.4	25.4
4	—	24.5	24.5
5	33.0 (c)	33.7 (n)	33.7 (n)
6	173.2 (c)	173.5	—
7	21.6	21.6	—
8	69.8	70.0	69.7
9	—	58.2	58.3
ACβ	ca.20 (β*),17.1 (rc)	ca.20 (β*),16.8 (rc+hrc)	16.7 (hrc)
ACα	49.3 (β*),50.0 (rc)	50.0 (rc+hrc)	50.0 (hrc)
GCα	42.7	42.7	42.7
SCβ	—, 61.4 (rc)	—, 61.4 (rc+hrc)	61.4 (hrc)
SCα	54.3 (β*),55.8(rc)	55.8(rc+hrc)	55.8 (hrc)

(c) and (n) indicate the crystalline and non-crystalline regions in PCL in PU. β*: anti-parallel β-sheet, rc: random coil, hrc: hydrated random coil. The numbers of carbon indicate the PU peaks noted in Figure 1.

**Table 2 molecules-26-04649-t002:** ^13^C chemical shifts (ppm) of ^13^C CP/MAS, ^13^C DD/MAS, and ^13^C r-INEPT NMR spectra of Sample III in the hydrated state together with the assignments.

Number of Carbons	CP/MAS	DD/MAS	INEPT
1	64.8 (c)	63.8 (n)	63.8 (n)
2	28.5 (c)	28.5 (n)	28.3 (n)
3	25.4	25.4	25.4
4	—	24.7	24.4
5	33.0 (c)	33.5 (n)	33.6 (n)
6	173.2 (c)	172.9	—
7	21.6	21.6	21.5
8	69.7	69.8	69.7
9	—	—	58.2
ACβ	ca.20 (β*),16.8 (rc)	ca.20 (β*),16.8 (rc+hrc)	16.7 (hrc)
ACα	49.4 (β*),50.0 (rc)	50.0 (rc+hrc)	50.0 (hrc)
GCα	42.7	42.7	42.7
SCβ	—,61.4 (rc)	—, 61.4 (rc+hrc)	61.4 (hrc)
SCα	54.3 (β*),55.8(rc)	55.8(rc+hrc)	55.8 (hrc)

(c) and (n) indicate the crystalline and non-crystalline regions in PCL in PU. β*: anti-parallel β-sheet, rc: random coil, hrc: hydrated random coil. The numbers of carbons indicate the PU peaks noted in Figure 1.

**Table 3 molecules-26-04649-t003:** The breaking strength (N) and elongation-at break (%) calculated from the stress-strain curves (Figure 8) of hydrated 3.5mm diameter SF knitted tubes without and with PU-SF composite sponge coating.

(a) Breaking strength (N)		
**n**	**SF tube without coating**	**SF tube with coating**
1	30.5	34.4
2	30.3	37.9
3	27.9	40.4
4	29.1	37.5
5	28.3	31.9
6	28.2	30.8
Average	29.1	35.5
Standard deviation	1.1	3.8
(b) Elongation-at-break (%)		
**n**	**SF tube without coating**	**SF tube with coating**
1	154	122
2	161	114
3	145	126
4	129	133
5	148	107
6	142	116
Average	147	120
Standard deviation	11	9

## Data Availability

Data is contained within the article.

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
