# Peer review of "Characterization of a Water-Dispersed Biodegradable Polyurethane-Silk Composite Sponge Using 13C Solid-State Nuclear Magnetic Resonance as Coating Material for Silk Vascular Grafts with Small Diameters"

_molecules, 2021, doi:10.3390/molecules26154649_

Round 1

Reviewer 1 Report

This manuscript describes the properties of knitted SF vascular graft coated with the water-dispersed biodegradable polyurethane-silk composite sponge, and in vivo experiments were also performed.  The introductory part of this manuscript is well written and the purpose of the study is also clearly presented. Since the results clearly show the high patency rate and good vascular endothelial cell formation within 4 weeks, it seems to be suitable for application to small-diameter vascular graft.  Therefore, this manuscript is recommended to be published in Molecules. I would like to ask you to answer some of questions.

  1. I want to know the reason why you autoclave SF graft after immersing in distilled water. Are there any problems with shape change or dissolution of materials during autoclaving?
  2. In Figure 9, unlike PU-SF coated SF graft, PU coated graft showed narrowing lumen diameter. Also, PU coated graft didn’t show the sufficient tissue infiltration. The reasons for the difference between the two is required to be clearly discussed. Is it because of difference in the degradation rate, inflammatory reaction, or biocompatibility of materials? If it is related to porosity, this should be confirmed either by measuring porosity or by observing the surface through SEM.

Reviewer 2 Report

Asakura and co-workers developed a water-dispersed biodegradable polyurethane-silk composite sponge as coating material for silk vascular grafts, and employed unique 13C solid-state NMR as an efficient tool for the characterization of the polymer structures and organization in the composites materials. The authors performed complete in vivo studies based on the degradable implants and the materials showed good performance in terms of biocompatibility and patency rate. I recommend its publication in Molecules, and here are the comments:

  1. The authors indicated that another paper highly relevant to this paper is still in submission state. I assume the characterization of the synthesis PU polymers was included in the other paper. The authors should clarify what kind of data have been divided, for example, 1H NMR, GPC and FTIR.

  1. The authors only showed partial structures of the polymers. It would be clearer to give all chemical structures of the monomers. In addition, it would be easier to check the 13C NMR signals if the chemical structures of the different groups can be directly placed in the NMR spectra.

  1. Regarding the difference in semicrystallinity of different PCL samples, would DSC or powder XRD can be used as supportive characterization tools?

  1. In figure 6, it’s hard to observe the difference of the sponge in 8 weeks and 6 months. How to tell the film is much thinner in 6 months?

  1. Why is that mild blood leakage was only observed for PU-SF-coating SF graft?

  1. 3D printing has become a powerful technique for the fabrication of biodegradable implants, including the stent-like devices (for example, Fisher et al, Adv. Mater. 2015, 27, 138–144; Leroux et al, Science Advances, 2021, 7, eabe9499; Park et al, Nature Communications 2018, 9, 1620, etc). A lot of advanced work have been done, the authors may consider to discuss this in the perspective part.

  1. Language:

Line 147, “biodegradable experiment” should be “in vivo degradation test”?

Line 157, “Preparation of SF Vascular Grafts Coated with a Water-Dispersed SF-PU Composite Sponge and Water-Dispersed PU Sponge Only”, this can cause confusion, the authors may give two abbreviations for easy understanding

Line 326, “contrary to” should be “in contrary to”

Line 335, “thus seem more suitable PU-SF composite 335 sponge for coating knitted SF grafts”, please rephrase

Line 354, “the swelling of the sponge became invisible to”, means “the sponge became invisible to”?

Please carefully check the expression to prevent misunderstanding

Author Response

"Please see attachment"

Reviewer 3 Report

Here the authors suggest a promising material (water-dispersed, biodegradable polyurethane-silk composite sponge) as a coating for small-diameter vascular grafts than can be employed in arterial reconstruction, bypass surgery, and microsurgery. The clinical relevance of such commercially available off-the-shelf medical devices is undoubted and any investigations in this direction deserve attention of the scientific and industrial communities. The authors performed an impressive chemical characterisation of their composite and attempted performing of a translational study to further show the perspectives of using this material in the routine clinical practice.

Although the paper proposes a technically novel approach for the construction of small-diameter vascular grafts which still represent an unmet clinical need in cardiovascular surgery, it has a number of major drawbacks precluding its further consideration for publication:

  1. Rat model, as well as other small animal models, is currently considered inappropriate for testing of small-diameter vascular grafts because biodegradation profile of the implants and hemostasis there are vastly different from those in large animals (e.g., sheep) and, expectedly, from those in humans.
  2. 4-week time point is insufficient for the proper assessment of primary patency. The observation period should not be less than 1 year. Longitudinal analysis must include both short-, mid-, and long-term time points to evaluate the frequency and mechanisms of thrombosis, neointimal hyperplasia, and calcification / degradation of the extracellular matrix.
  3. Please carry out an ultrastructural investigation of the grafts and sponges, e.g. by means of scanning electron microscopy. Such imaging may also be repeated upon the implantation to assess the integrity of the endothelial layer and structural heterogeneity of the grafts at ascending time points.
  4. Tensile testing (i.e., quantitative evaluation of yield and ultimate tensile stress, elasticity, stiffness and calculation of the stress-strain curve)  is mandatory for any assessment of small-diameter vascular grafts. Thickness of such scaffolds and its variability across the samples should also be indicated. Importantly, all these parameters must be evaluated upon the sterilisation to reproduce the pre-implantation scenario.
  5. The experiments do not provide any insight into the mechanisms of biocompatibility. Without immunophenotyping, it is barely possible to analyze the cellular composition of the vascular tissue replacing the polymer scaffold and whether it corresponds to that of native blood vessels. 
  6. Time-resolved molecular analysis of genomic and proteomic signatures (e.g., by means of qPCR or Western blotting) of vascular regeneration at the site of implantation would be also beneficial for the paper. This can be achieved by homogenisation of the excised tissues at ascending time points, with subsequent RNA and total protein extraction.

Minor corrections

  • please enlarge the scale bars at histological images to make them clearly visible at all figures and for all readers
  • when performing any quantitative analysis in the future, please design all graphs as box-and-whisker plots and indicate all replicates as points at the graph (e.g., box-and-whisker, min to max, show all points if using GraphPad Prism).

To summarise, the paper describes an initial stage of the development of small-diameter vascular graft rather than a complete or near-complete testing. The methodology must conform to the standards of the field before the paper can be processed further.

Author Response

Response to Reviewer 3:

We wish to express our appreciation to the reviewer for their insightful comments on our paper. The comments have helped us significantly improve the paper.

Comment 1: Rat model, as well as other small animal models, is currently considered inappropriate for testing of small-diameter vascular grafts because biodegradation profile of the implants and hemostasis there are vastly different from those in large animals (e.g., sheep) and, expectedly, from those in humans.

Response: Our work is a basic research to develop small-diameter silk vascular grafts submitted to “molecules”. Therefore, in the future, it will be necessary to implant the silk grafts into large animals for investigation patency and remodeling ability.

We have therefore changed the following text from (p. 19, lines 510):

“In the future, we would like to examine the optimum amount of PU to be mixed with SF by conducting physical property tests of the created SF grafts.”

to

In the future, we would like to examine the optimum amount of PU to be mixed with SF by conducting implant PU-SF-coated SF grafts into large animals and investigate patency and remodeling ability for practical application.

Comment 2: 4-week time point is insufficient for the proper assessment of primary patency. The observation period should not be less than 1 year. Longitudinal analysis must include both short-, mid-, and long-term time points to evaluate the frequency and mechanisms of thrombosis, neointimal hyperplasia, and calcification / degradation of the extracellular matrix.

Response: As reviewer pointed out, 4-week time point might be insufficient. Our research group has already conducted transplantation experiments on artificial SF vascular grafts using rats. In the first implanted study, we observed the patency and remodeling ability at 2 weeks, 12 weeks, and 1 year after implantation. It was found that SF fibers are gradually degraded and replaced with self-tissue without causing intimal hyperplasia and calcification (reference 18). Since vascular endothelial cells need to cover grafts for long-term patency, we have continued to improve remodeling ability of graft after implantation. In our previous studies, vascular endothelial cells could not be confirmed 2 weeks after implantation. On the other hand, 3 months after implantation, vascular endothelial cells completely covered the grafts, and remodeling was completed (reference 32). Therefore, it was important to confirm the appearance of vascular endothelial cells at what point after implantation. For the above reasons, an observation point was set up 4 weeks after implantation in this study. However, since the coating method used in this study is an improvement of the conventional coating, we would like to evaluate it not only 4 weeks after implantation but also in the medium to long term in the future.

We have therefore added the following text as one of the limitations on this study (p. 19, lines 510):

In the future, we would like to examine the optimum amount of PU to be mixed with SF by conducting implant PU-SF-coated SF grafts into large animals and investigate patency and remodeling ability for practical application.

Comment 3: Please carry out an ultrastructural investigation of the grafts and sponges, e.g. by means of scanning electron microscopy. Such imaging may also be repeated upon the implantation to assess the integrity of the endothelial layer and structural heterogeneity of the grafts at ascending time points.

Response: We prepared the PU-SF sponge and PU-SF coated SF graft again for ultrastructural investigation by means of SEM. The SEM pictures are included in the text as new Figure 7. The 50-100 mm holes in the inner part of the sponge were observed in Figure 7 (a). Figure 7(b) showed that the space among woven silk fibers of the grafts was lightly covered with the PU-SF sponge.

Comment 4: Tensile testing (i.e., quantitative evaluation of yield and ultimate tensile stress, elasticity, stiffness and calculation of the stress-strain curve) is mandatory for any assessment of small-diameter vascular grafts. Thickness of such scaffolds and its variability across the samples should also be indicated. Importantly, all these parameters must be evaluated upon the sterilization to reproduce the pre-implantation scenario.

Response: Because the diameter of 1.5 mm of the SF double-raschel knitted tube is too small for tensile testing, we used SF double-raschel knitted tube with 3.5mm diameter prepared by the same knitting matter as that of 1.5 mm diameter. The tensile testing of the SF double-raschel knitted tube coated with PU-SF composite sponge and SF double-raschel knitted tube without coating was performed in the hydrated state using an EZ-Graph tensile testing machine (EZ-Graph, SHIMADZU Co. Ltd. Japan) at room temperature according the methods reported previously [32,36]. Namely, the samples were immersed in distilled water for 24 hrs and then the stress-strain curve was observed immediately after wiping the water on the surface of the samples. The breaking strength (N) and elongation-at break (%) was calculated from the stress-strain curves (n =6). The thickness and the standard deviation (n=10 each) of the hydrated SF double-raschel knitted tubes without and with PU-SF composite sponge coating was determined using micrometer (MDC-25PJ, Mitutoyo Co. Japan).

Comment 5: The experiments do not provide any insight into the mechanisms of biocompatibility. Without immunophenotyping, it is barely possible to analyze the cellular composition of the vascular tissue replacing the polymer scaffold and whether it corresponds to that of native blood vessels. 

Response: In contrast to non-natural polymers, silk can be remodeled allowing cell attachment and infiltration which leads to degradation of the scaffold and replacement by native tissue over time. Thus, the silk scaffold provided a material in which the cells can self-organize into vessel-like structures. Therefore, in this study, immunohistochemical staining was performed to prove whether the blood vessels were remodeled by self-tissue, and also commonly used in many past development studies of small-diameter artificial vascular grafts.

Comment 6: Time-resolved molecular analysis of genomic and proteomic signatures (e.g., by means of qPCR or Western blotting) of vascular regeneration at the site of implantation would be also beneficial for the paper. This can be achieved by homogenisation of the excised tissues at ascending time points, with subsequent RNA and total protein extraction.

Response: We agree with the reviewer that additional analysis would be beneficial for this paper. However, our purpose in this study is to prepare a small-diameter artificial vascular graft in which PU is mixed with silk fibroin, characterize it, and evaluate it in vivo after implantation. Therefore, it was not possible to increase the reviewer recommended analysis in this study.

Comment 7: please enlarge the scale bars at histological images to make them clearly visible at all figures and for all readers.

Response: In accordance with the reviewer's comment, we have enlarged the scale bars at previous Figure 8 and 9.

Thank you again for your comments on your paper. We trust that the revised manuscript is suitable for publication.

Round 2

Reviewer 3 Report

I am still convinced that small animal models are nonsensical at best and misleading at worst for the testing of tissue-engineered vascular grafts. Therefore, I cannot see any justification to publish a work where the authors describe the implantation of another vascular graft prototype into small animals. Further, the experiment is limited to a short-term implantation.

Revised Figure 7 does not show the appearance of the graft post implantation. The tensile testing should be analyzed comprehensively, including statistically meaningful comparison of all parameters (yield and ultimate tensile strength, elongation at break and elastic modulus). That is, P values must be added to the Revised Table S1. Immunohistochemical analysis from Revised Figure 11 needs quantitation.